# Agronomic Performance of RR® Soybean Submitted to Glyphosate Application Associated with a Product Based on *Bacillus subtilis*

Guilherme Braga Pereira Braz [1,*] , Eduardo Souza Freire [1] , Bruno César Silva Pereira [1] , Fernanda dos Santos Farnese [2] , Matheus de Freitas Souza [1] , Lucas Loram-Lourenço [2] and Letícia Ferreira de Sousa [2]

[1] Agronomy Department, Universidade de Rio Verde, Rio Verde 75901-970, Brazil
[2] Agricultural Sciences Center, Instituto Federal Goiano, Rio Verde 75901-970, Brazil
* Correspondence: guilhermebrag@gmail.com; Tel.: +55-064-3611-2200

**Abstract:** Despite the great benefits arising from the adoption of Roundup Ready® (RR®) soybean, there are reports about the lack of selectivity of glyphosate for this crop. The use of growth-promoting microorganisms can help attenuate the injuries caused by herbicides. The objective of this work was to evaluate the agronomic performance of RR® soybean submitted to the post-emergence application of glyphosate both isolated and in association with *Bacillus subtilis*. The experiment was carried out in a completely randomized block design, with four replications. The treatments consisted of the post-emergence applications of glyphosate (1296 g a.i. ha$^{-1}$), glyphosate (2592 g a.i. ha$^{-1}$), glyphosate/glyphosate (1296/1296 g a.i. ha$^{-1}$), glyphosate + *B. subtilis* BV02 (1296 + 42 g a.i. ha$^{-1}$), glyphosate + *B. subtilis* BV02 (2592 + 42 g a.i. ha$^{-1}$), and glyphosate + *B. subtilis* BV02/glyphosate + *B. subtilis* BV02 (1296 + 42/1296 + 42 g a.i. ha$^{-1}$). The application of glyphosate (2592 g a.i. ha$^{-1}$) and the sequential application of glyphosate provides higher levels of intoxication. The association of *B. subtilis* BV02 with glyphosate (2592 g a.i. ha$^{-1}$) prevented losses in the values of relative chlorophyll *a* and *b* and the total chlorophyll index. The soybean yield was reduced when the plants were submitted to a sequential application of glyphosate.

**Keywords:** EPSPs inhibitors; *Glycine max*; growth-promoting microorganisms; selectivity

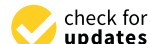



## 1. Introduction

Since the introduction to the market of genetically modified soybean cultivars that are tolerant to the application of the herbicide glyphosate, a technology known as Roundup Ready® (RR®), profound changes have been seen in the weed management of this crop. The possibility of using glyphosate in post-emergence soybean applications occurred through the insertion of a gene that encodes the EPSPS enzyme, causing the plants of this crop to present an enzyme insensitive to that herbicide [1].

The profound changes in the weed management of soybeans mentioned above refer to the flexibility in the stage of the crop plants regarding the application of glyphosate, in addition to the fact that this active ingredient is characterized as having a broad spectrum, that is, controls monocots (grasses) and dicots (broadleaves). Further, glyphosate has a high translocation after absorption (systemic), which creates the possibility of controlling weeds in advanced stages, also because glyphosate has no residual activity in the soil, eliminating carryover risks under crops to be implemented in succession [2]. Proof of the great benefits inherent in the use of RR® cultivars can be seen by the wide acceptance by farmers, since, at present, ≈92% of the area cultivated with soybean in Brazil is sown with transgenic cultivars containing tolerance to glyphosate [3].

Despite the great benefits arising from the adoption of RR® soybeans, since the introduction of these transgenic cultivars in production systems, reports regarding the lack of

selectivity of glyphosate for this crop are frequent, and in certain situations, the occurrence of yield reductions is pointed out. Several studies have already been published in the literature to present the deleterious effect that glyphosate has on soybean. It has shown negative effects on morphological (plant height), physiological (water use efficiency), and yield (yield components) parameters [4–6].

However, it is worth noting that despite the negative effects of glyphosate on certain growth and production parameters of soybeans, when the correct positioning of the application of this herbicide is respected, sometimes, there is no great damage to the crop development [2]. However, for situations in which there is a chance of low selectivity resulting from the use of glyphosate in soybeans, such as the application of doses beyond those recommended in the label insert, the use of glyphosate in sequential applications, and the application of this active ingredient in plants under abiotic stress (e.g., drought), it is sometimes necessary to adopt measures to mitigate the deleterious effects of glyphosate on the crop.

In this sense, aiming at mitigating the deleterious effects resulting from the post-emergence application of glyphosate in RR® soybean cultivars, one possibility that demonstrates the feasibility of use refers to the application of product classes that provide better vegetative and reproductive development for plants, such as foliar fertilizers, biostimulants, and amino acids [6,7]. To mitigate the effects caused by the lower selectivity of glyphosate to RR® soybeans, it is necessary to carry out studies on the use of microorganisms that promote plant growth (biocontrol agents) in the prevention of injuries caused by this active ingredient.

To elucidate more clearly why it is important to develop studies that evaluate the potential of the exploitation of products used in biological control for mitigating the deleterious effects of glyphosate, it was seen in a work described in the literature that plants that had their seeds treated with *Bacillus amyloliquefaciens* showed greater development of the root system and better architecture [8]. One of the mechanisms of the detoxification of herbicides by plants is related to the metabolization of these products into inactive compounds, a mechanism that has a high-water demand to be exercised by plants [4–6]. In this sense, plants with a more robust root system will be able to exploit a greater volume of soil for water absorption, helping the detoxification process of herbicides in situations in which these products were applied to plants subjected to a water deficit [6,7,9].

Given the context discussed above, the objective of this study was to evaluate the agronomic performance of the RR® soybean submitted to the post-emergence application of glyphosate both isolated and in association with a product based on *Bacillus subtilis* BV02 at different stages and doses.

## 2. Materials and Methods

The experiment was carried out at an experimental area located in the municipality of Rio Verde (Brazil), from 20 November 2020 to 12 March 2021. This experiment was specifically installed at latitude 17°47′15.03″ S and longitude 50°57′39.46″ W, at 767 m of altitude.

The soil of the experimental area was classified as *Latossolo Vermelho distroférrico* (Rhodic Ferralsol) [10]. Before the installation of the experiment, soil sampling was carried out at a depth of 0 to 20 cm, which revealed the following physicochemical properties: pH in $CaCl_2$ of 5.3, organic matter content of 37.6 g $dm^{-3}$, and sand, silt, and clay contents of 49.5, 2.0, and 48.5%, respectively (clay texture). According to the Köppen classification, the climate of the municipality of Rio Verde is of the Aw type, which is called "tropical with a dry season", characterized by having more intense rains in the summer compared to winter. Climatological data related to maximum and minimum temperature, relative air humidity, and rainfall during the experiment conduction period are shown in Figure 1.

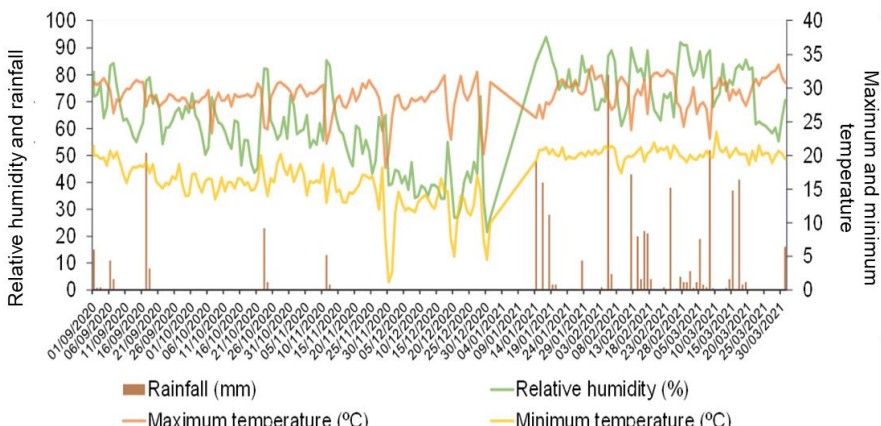

**Figure 1.** Maximum and minimum temperature, relative humidity, and rainfall observed during the experiment carried out with soybean submitted to post-emergence application of glyphosate isolated and associated with *B. subtilis* BV02. Source: Instituto Nacional de Meteorologia. Rio Verde (Brazil).

Before sowing, the burndown was performed using two herbicide applications, the first being carried out fifteen days before sowing with the application of the association between glyphosate plus clethodim (1620 + 80 g a.i. ha$^{-1}$), and the second carried out on the day of sowing, with the application of the herbicide glufosinate (500 g a.i. ha$^{-1}$) with the addition of vegetal oil (Aureo®, Bayer CropScience, Leverkusen, Germany) at 0.5% v v$^{-1}$. Sowing was carried out on 20 November 2020, in a direct sowing system, adopting 0.5 m spacing between rows. Eighteen seeds were distributed per linear meter of the soybean cultivar NS 7709 IPRO® (NIDERA Seeds, São Paulo, Brazil), which has an indeterminate growth habit and maturation group 7.2. The seeds used in the experiment received industrial treatment with fungicides and insecticides. The fertilization of the crop was carried out at the time of sowing with 350 kg ha$^{-1}$ of 02-20-18 (N-P-K). The emergence of soybean seedlings occurred on 25 November 2020.

The experimental design used was completely randomized blocks, in a double-adjacent check scheme, adopting four replications. In this experimental arrangement, each evaluated treatment has two adjacent checks, which are used to compare the values obtained for each variable. The adoption of this experimental arrangement allows for the comparison (splitting) of the herbicide treatments with the controls that were installed within the same plot, which consequently minimizes the variability of the area and the experimental error with efficiency, which is fundamental for experiments that evaluate the herbicide selectivity [11].

Table 1 shows the treatments that were evaluated in the present experiment, along with their respective doses and stage of application. The commercial product used based on glyphosate was Roundup Transorb® (isopropylamine salt), which presented a concentration of 648 g L$^{-1}$ of active ingredient (a.i.), which is equivalent to 480 g L$^{-1}$ of acid equivalent (a.e.). To evaluate the attenuation potential of a product based on *Bacillus subtilis* BV02, the commercial product selected was Bio-Imune® (Grupo Vittia, São Joaquim da Barra, Brazil). The strain of this bacteria is originated from Brazil. The experimental units consisted of eight sowing lines, spaced 0.5 m apart, with a length of 5.0 m (20.0 m$^2$). Only the four central lines of the experimental unit were considered as a useful area for the evaluations, excluding 0.5 m from each end, totaling an area equivalent to 8.0 m$^2$.

To ensure that soybean plants were only exposed to the effects of herbicide treatments, weeding of the species that made up the weed community of all experimental units was carried out throughout the entire crop cycle. In addition, during soybean development, cultural treatments were carried out following the technical recommendations [12], controlling pests and diseases without allowing them to negatively influence crop development. All applications of insecticides and fungicides were carried out using a sprayer, adopting a spray volume equivalent to 150 L ha$^{-1}$.

**Table 1.** Treatments and respective doses used in the experiment carried out with soybean submitted to post-emergence application of glyphosate at different doses and stages and associated with *B. subtilis* BV02.

| Treatments | Dose (g a.i. ha$^{-1}$) | Phenological Stage |
|---|---|---|
| Glyphosate | 1296 | V4 to V5 |
| Glyphosate | 2592 | V4 to V5 |
| Glyphosate/glyphosate | 1296/1296 | V4 to V5/V5 to V6 |
| Glyphosate + *B. subtilis* BV02 | 1296 + 42 | V4 to V5 |
| Glyphosate + *B. subtilis* BV02 | 2592 + 42 | V4 to V5 |
| Glyphosate + *B. subtilis* BV02/Glyphosate + *B. subtilis* BV02 | 1296 + 42/1296 + 42 | V4 to V5/V5 to V6 |

"/" = sequential application; "+" = tank mixture. Silkon® was added in all treatments at 1.0 L c.p. ha$^{-1}$ dose.

The first application (Application A) was carried out on 21 December 2020 (4:30 pm to 5:30 pm), which was carried out in all treatments. On this occasion, the soybean plants were in the V4-V5 stage and had heights varying between 30 and 35 cm. At the time of application, the soil was humid, the air temperature was 29.0 °C, the relative humidity was 35%, with cloudy skies and winds with a speed of 2.0 km h$^{-1}$. On 04 January 2021 (5:10 pm to 5:25 pm) the sequential application was carried out 14 days after Application A (DAA-A), which included only the treatments provided in Table 1. The soybean plants were in the V5-V6 stage, and the plants had an average height of 35 to 40 cm. At the time of application, the soil was humid, the air temperature was 26.0 °C, the relative humidity was 62%, the sky was cloudy, and the wind was 0.9 km h$^{-1}$. In all treatment applications, a $CO_2$-based constant pressure backpack sprayer was used, equipped with a bar with five fan tips XR-110.02, under a pressure of 2.0 kgf cm$^{-2}$. These application conditions provided the equivalent of 200 L ha$^{-1}$ of spray solution.

To measure the effect of treatments on the soybean, evaluations of the percentage of phytotoxicity were carried out, using a visual scale proposed by the SBCPD [13], in which 0% means the absence of symptoms and 100% total plant death, with evaluations performed at 7, 14, 21, and 35 DAA-A. Evaluations were also carried out regarding the Relative Chlorophyll Index (CRI) using the ClorofiLOG® device manufactured by the company Falker® (FALKER, Porto Alegre, Brazil). To measure the values of chlorophyll a, b, and total chlorophyll, as well as the ratio between chlorophyll a and b, the evaluation was carried out in the second trifoliate leaf completely expanded from the apex to the base of the soybean plants, being the measurement performed in the central leaflet. For this evaluation, five plants were sampled per experimental unit, and measurements were taken at 21 DAA-A.

Furthermore, evaluations were carried out to determine the effect of treatments on the physiological parameters of soybean plants, with measurements being performed at 2 DAA-A and 2 days after the second application (DAA-B). To assess the occurrence of membrane damage, electrolyte leakage was determined in ten leaf discs with a diameter of 1 cm, placed in a test tube containing 10 mL of deionized water, and kept in a water bath at 25 °C for 24 h. Then, the electrical conductivity of the solution (L1) was read. Subsequently, the test tubes were incubated at 90 °C for 1 h, and after reaching thermal equilibrium with the environment, a new reading (L2) of the electrical conductivity of the solution was carried out. Electrolyte leakage was calculated according to the following equation [14].

$$EL (\%) = (L1/L2) * 100$$

Chlorophyll *a* fluorescence parameters were measured in all plants, and the minimum fluorescence ($F_0$) was obtained in the morning via excitation of leaf tissues by low-intensity modulated red light (0.03 μmol photons m$^{-2}$ s$^{-1}$). Maximum fluorescence (Fm) was obtained by applying a 0.8 s pulse of saturating actinic light (8000 μmol photons m$^{-2}$ s$^{-1}$). The variable fluorescence (Fv) was determined by the difference between $F_0$ and Fm, and from these values, the potential quantum yield of photosystem II was calculated. The

leaves were acclimated to actinic light (1000 μmol photons m$^{-2}$ s$^{-1}$) for 60 s to obtain transient fluorescence (Fs), followed by a saturating light pulse to estimate the maximum fluorescence at the light (Fm′), and, finally, a far-red light pulse was applied to obtain the minimum fluorescence after acclimation to actinic light (F$_0$′). With these parameters, the photochemical extinction coefficients (qP), the photochemical efficiency of electron transport associated with photosystem II (φFSII), and the non-photochemical quenching (NPQ) were calculated [15].

The net carbon assimilation rate (A), stomatal conductance (gs), internal $CO_2$ concentration (Ci), and transpiration rate (E) were determined in an open system, under saturating light (1000 μmol m$^{-2}$ s$^{-1}$) and $CO_2$ partial pressure of 40 Pa. For this purpose, an infrared gas analyzer (LI-6400, Li-Cor Inc., Lincoln, NE, USA) equipped with a blue/red light source (model LI-6400-02B, LI-COR) was used. With the same analyzer, nocturnal respiration, or the nocturnal net $CO_2$ assimilation rate, was evaluated before dawn.

In addition, at the time of the soybean harvest, the stand and height of plants were measured. In the evaluation of the stand, the number of plants present in 2 linear m of the useful area of each experimental unit was counted. To evaluate the height of plants, the distance from the ground level to the insertion of the last fully expanded trefoil of soybean plants was performed, sampling five plants per experimental unit. Furthermore, the number of pods per plant was evaluated, counting the total number of pods in five plants randomly collected in the useful area of the experimental units. To determine yield grains, on 12 March 2021, all plants present in the useful area of each experimental unit were manually harvested, which were subsequently threshed, packed, identified, and weighed, and the grain moisture was corrected to 13.0% in all treatments and checks. The results of the grain yield were expressed in percentage related to the check.

Data analysis was performed using the SISVAR software (5.6 version) [16]. For the statistical analysis of phytotoxicity data and the variables related to photosynthesis, the data were submitted for analysis of variance by the F test, and the means were compared by the SNK test ($p \leq 0.05$). For the other variables, an analysis of variance was performed, and later, a comparison of the splitting of the herbicide treatment with the respective double check was compared by the F test ($p \leq 0.05$).

## 3. Results

Analyzing the levels of intoxication among all evaluations (7, 14, 21, and 35 DAA-A), it appears that the symptoms seen after the application of glyphosate were characterized as mild (≤15.00%) (Table 2). Regarding the symptoms, yellowing can be observed in the soybean trifoliate leaves, and visually, in the treatments with higher doses, apparently the plants had a slight reduction in the intensity of closure between the rows (canopy). In the first evaluation (7 DAA-A), the highest percentages of injuries to soybean plants were observed in treatments containing glyphosate application at doses equivalent to 2592 g a.i. ha$^{-1}$, regardless of the status of association with *B. subtilis* BV02. On this occasion, the levels of phytotoxicity did not exceed the level of 10.00%, being characterized as low-intensity injuries. In this evaluation (7 DAA-A) and the following one (14 DAA-A), the sequential application had not yet been carried out in the treatments in which this operation was foreseen, and the symptoms observed came only from the first application.

At 14 DAA-A, the highest percentage of intoxication in the soybean plants was observed in the treatment with the isolated application of glyphosate (2592 g a.i. ha$^{-1}$). This result indicates that the addition of *B. subtilis* BV02 in association with glyphosate (2592 g a.i. ha$^{-1}$) was able to provide slight attenuation in the symptoms of intoxication in the soybean plants. In contrast, in the evaluation carried out at 21 DAA-A, the levels of intoxication of the soybean plants were similar, regardless of the association of *B. subtilis* BV02 with glyphosate. On this occasion, the most pronounced symptoms of injuries were observed in the treatments with a sequential application of glyphosate (1296 g a.i. ha$^{-1}$), followed by glyphosate at a dose of 2592 g a.i. ha$^{-1}$ and glyphosate at the lowest dose (1296 g a.i. ha$^{-1}$).

In the final phytotoxicity assessment, carried out at 35 DAA-A, the results were similar to those observed at 21 DAA-A, with higher levels of injuries being seen in soybean plants when they were submitted to the application of higher doses. Table 3 presents the results of the evaluations related to the relative chlorophyll index, in which the variables chlorophyll a, b, total chlorophyll, and the ratio between chlorophyll a and b were measured. Excluding the ratio between chlorophyll *a* and *b*, for all others, differences between treatments and their respective double checks were observed.

**Table 2.** Soybean phytotoxicity after post-emergence application of glyphosate at different doses and stages and association with *B. subtilis* BV02.

| Treatments | Dose (g a.i. ha$^{-1}$) | Phytotoxicity (%) | | | |
|---|---|---|---|---|---|
| | | 7 DAA-A | 14 DAA-A | 21 DAA-A | 35 DAA-A |
| Glyphosate | 1296 | 6.25 b | 5.75 b | 1.25 c | 0.00 c |
| Glyphosate | 2592 | 10.00 a | 9.50 a | 8.50 b | 6.50 b |
| Glyphosate/glyphosate | 1296/1296 | 5.75 b | 3.75 b | 13.50 a | 11.00 a |
| Glyphosate + *B. subtilis* | 1296 + 42 | 6.50 b | 4.50 b | 0.00 c | 0.00 c |
| Glyphosate + *B. subtilis* | 2592 + 42 | 10.00 a | 6.50 b | 6.50 b | 5.00 b |
| Glyphosate + *B. subtilis*/Glyphosate + *B. subtilis* | 1296 + 42/1296 + 42 | 6.50 b | 5.75 b | 14.25 a | 10.50 a |
| CV (%) | | 21.08 | 29.00 | 21.94 | 16.03 |

Means followed by the same letter do not differ from each other by the SNK test ($p \leq 0.05$).

**Table 3.** Relative chlorophyll index in soybean plants submitted to post-emergence application of glyphosate at different doses and stages and association with *B. subtilis* BV02.

| Treatments | Dose (g a.i. ha$^{-1}$) | Chlorophyll *a* | | Chlorophyll *b* | |
|---|---|---|---|---|---|
| | | Herbicide | DC | Herbicide | DC |
| Glyphosate | 1296 | 30.82 a | 31.62 a | 7.62 a | 9.82 a |
| Glyphosate | 2592 | 26.81 b | 32.04 a | 6.93 b | 9.80 a |
| Glyphosate/glyphosate | 1296/1296 | 24.95 b | 33.76 a | 5.88 b | 8.46 a |
| Glyphosate + *B. subtilis* | 1296 + 42 | 30.76 a | 33.22 a | 7.46 a | 8.48 a |
| Glyphosate + *B. subtilis* | 2592 + 42 | 30.14 a | 30.97 a | 7.52 a | 8.10 a |
| Glyphosate + *B. subtilis*/Glyphosate + *B. subtilis* | 1296 + 42/1296 + 42 | 25.14 b | 31.42 a | 6.25 a | 7.89 a |
| CV (%) | | 9.07 | | 17.67 | |
| | | Total chlorophyll | | Ratio chlorophyll *a* and *b* | |
| | | Herbicide | DC | Herbicide | DC |
| Glyphosate | 1296 | 38.44 a | 41.44 a | 4.11 | 3.83 |
| Glyphosate | 2592 | 33.74 b | 41.85 a | 3.96 | 3.89 |
| Glyphosate/glyphosate | 1296/1296 | 30.83 b | 42.23 a | 4.29 | 4.16 |
| Glyphosate + *B. subtilis* | 1296 + 42 | 38.23 a | 41.70 a | 4.24 | 4.07 |
| Glyphosate + *B. subtilis* | 2592 + 42 | 37.66 a | 39.08 a | 4.13 | 3.93 |
| Glyphosate + *B. subtilis*/Glyphosate + *B. subtilis* | 1296 + 42/1296 + 42 | 31.39 b | 39.32 a | 4.07 | 4.13 |
| CV (%) | | 8.94 | | 7.88 | |

DC = double check. For each response variable, means followed by different letters in the line differ from each other by the F Test ($p \leq 0.05$).

For the evaluations of relative chlorophyll *a*, *b*, and total chlorophyll content, when soybean plants were subjected to the post-emergence application of higher doses of glyphosate,

regardless of whether the herbicide dose was used in a single or sequential application, there was a reduction in the values of these variables compared to the respective double checks. In contrast, for the relative index of chlorophyll *a*, *b*, and total chlorophyll, there was no reduction in the values of these variables compared to the double check when *B. subtilis* BV02 was added to the glyphosate application solution when the herbicide was applied at the highest dose (2592 g a.i. ha$^{-1}$). Furthermore, for the relative chlorophyll *b* index, no differences were observed in the values recorded for the double check and the treatment composed by the sequential application of the association between glyphosate plus *B. subtilis* BV02.

The observed decrease in total chlorophyll was not reflected in drops in photosynthetic rate (Figure 2A). None of the concentrations and forms of the isolated glyphosate application reduced the rate of carbon fixation. However, soybean plants submitted to a single application of glyphosate (2592 g a.i. ha$^{-1}$) or a sequential application of glyphosate (1296 g a.i. ha$^{-1}$), both in association with *B. subtilis* BV02 and without, showed an increase in photosynthesis compared to the check without the herbicide application. The higher carbon fixation was probably a consequence of the increase in stomatal conductance in the aforementioned treatments, and this result was also observed for the treatment composed of the single application of isolated glyphosate (1296 g a.i. ha$^{-1}$) (Figure 2B).

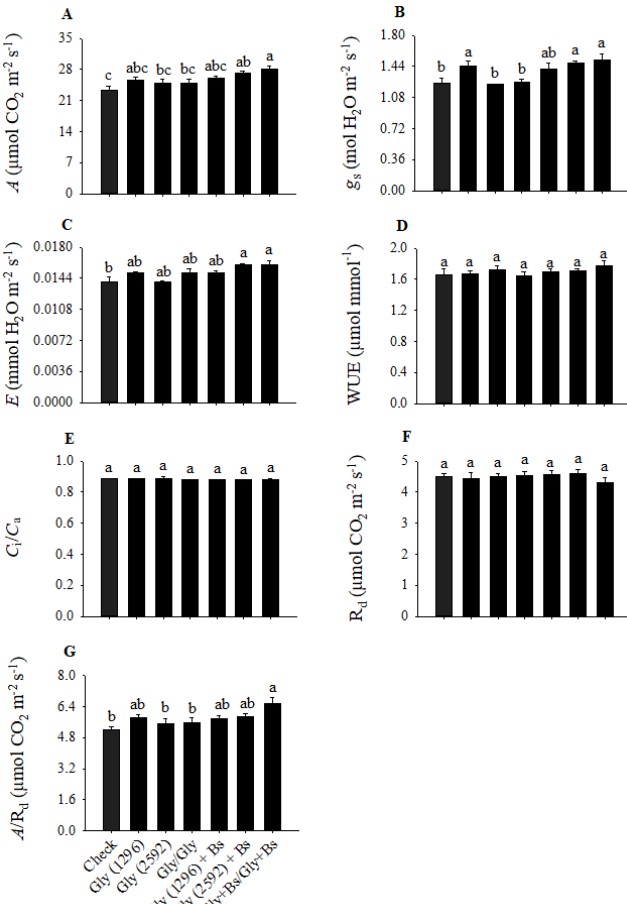

**Figure 2.** Gas exchange parameters in plants submitted to different doses of glyphosate, with or without application of *B. subtilis* BV02. The following parameters were evaluated: rate of net carbon assimilation ($A$-(**A**)), stomatal conductance ($g_s$-(**B**)), transpiration ($E$-(**C**)), water use efficiency (WUE-(**D**)), the ratio between internal and the external concentration of $CO_2$ ($C_i/C_a$-(**E**)), nocturnal respiration (Rd-(**F**)), and the ratio between photosynthetic rate and respiration (A/Rd-(**G**)). Means followed by the same lowercase letters do not differ according to the SNK test ($p \leq 0.05$).

The highest values of the transpiration rate were recorded in the soybean plants that received a single application of glyphosate (2592 g a.i. ha$^{-1}$) or a sequential application of this herbicide (1296 g a.i. ha$^{-1}$) in association with *B. subtilis* BV02. In these treatments, statistical differences were observed in relation to the check without the herbicide application (Figure 2C). The higher transpiration reflects the higher *g*s observed in these treatments, but these parameters did not influence the water use efficiency of the plants (Figure 2D). Similarly, the parameters of $C_i/C_a$ and nocturnal respiration also did not change in any of the treatments analyzed (Figure 2E,F). Regarding the ratio between the photosynthetic rate and nocturnal respiration, only the sequential treatment with glyphosate, associated with *B. subtilis* BV02, resulted in a significant increase, indicating a more positive carbon balance, which may result in gains in growth and yield. Regarding the chlorophyll *a* fluorescence parameters, it was observed that $F_v/F_m$, qP, and NPQ did not change in any of the treatments (Figure 3A–C). The sequential application of glyphosate plus *B. subtilis* BV02, however, increased the value of this variable compared to the check, which, associated with greater stomatal opening, may have contributed to the greater carbon fixation in this treatment (Figure 2D).

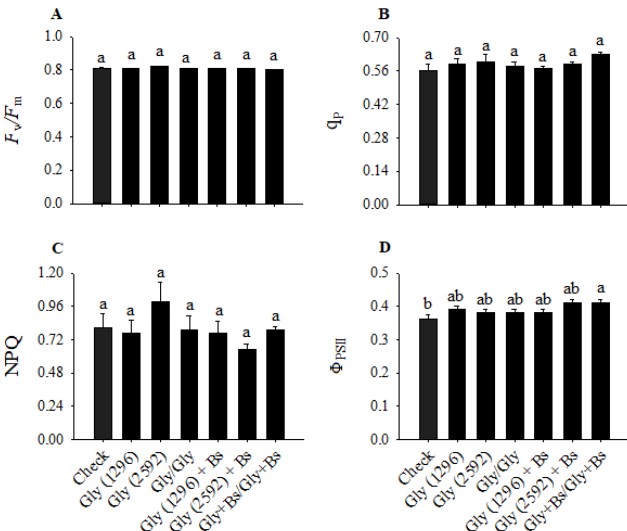

**Figure 3.** Chlorophyll *a* fluorescence parameters in plants submitted to different doses of glyphosate, with or without application of *B. subtilis* BV02. The following parameters were evaluated: potential quantum yield of photosystem II ($F_v/F_m$–(**A**)), the photochemical extinction coefficient (qP–(**B**)), the non-photochemical quenching (NPQ–(**C**)), and the effective quantum yield of photosystem II (ΦPSII–(**D**)). Means followed by the same lowercase letters do not differ according to the SNK test ($p \leq 0.05$).

Regarding the measurement of electrolyte extravasation, it appears that in the treatments with the application of isolated glyphosate (1296 g a.i. ha$^{-1}$), there was an increase in extravasated values compared to the check (Figure 4). In this sense, it is observed that when *B. subtilis* BV02 was added to the application of these treatments, no differences were observed in comparison with the check, a fact that demonstrates that the aforementioned microorganism may have acted in the process of attenuation of electrolyte leakage. For this variable, the application of glyphosate in the lower doses provided increases in the electrolyte leakage compared to the check. For the second evaluation of gas exchange parameters (2 DAA-B), no differences were observed in any of the variables analyzed (data not shown).

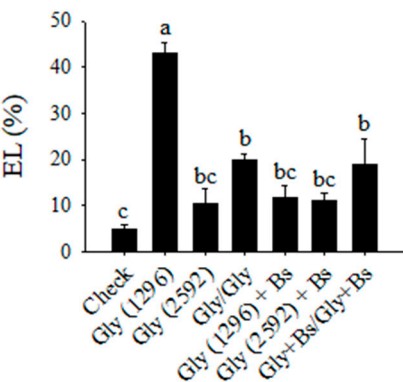

**Figure 4.** Electrolyte extravasation in plants submitted to different doses of glyphosate, with or without application of *B. subtilis* BV02. Means followed by the same lowercase letters do not differ according to the SNK test ($p \leq 0.05$).

Table 4 shows the results of evaluations of the stand and plant height of the soybean. Both for the evaluation of the crop stand, as well as for the plant height of the soybean, no differences were observed between treatments and their respective double checks. For the yield component number of pods per plant, there was no negative effect of treatments compared to the values observed in the respective checks without herbicides, demonstrating that glyphosate, regardless of the dose or number of applications, did not provide negative effects on the number of pods.

**Table 4.** Stand, plant height, number of pods, and soybean yield after post-emergence application of glyphosate at different doses and stages and the association with *B. subtilis* BV02.

| Treatments | Dose (g a.i. ha$^{-1}$) | Stand (pls 1 m$^{-1}$) | | Plant Height (cm) | |
|---|---|---|---|---|---|
| | | Herbicide | DC | Herbicide | DC |
| Glyphosate | 1296 | 14.57 | 14.77 | 112.50 | 118.25 |
| Glyphosate | 2592 | 14.47 | 14.67 | 116.00 | 119.50 |
| Glyphosate/glyphosate | 1296/1296 | 14.67 | 14.47 | 113.50 | 117.50 |
| Glyphosate + *B. subtilis* | 1296 + 42 | 14.57 | 14.82 | 123.75 | 118.50 |
| Glyphosate + *B. subtilis* | 2592 + 42 | 14.52 | 14.47 | 114.75 | 118.50 |
| Glyphosate + *B. subtilis*/Glyphosate + *B. subtilis* | 1296 + 42/1296 + 42 | 14.47 | 14.62 | 119.00 | 119.75 |
| CV (%) | | 3.12 | | 4.82 | |
| | | Pods per plant | | Relative yield (%) | |
| | | Herbicide | DC | Herbicide | DC |
| Glyphosate | 1296 | 51.7 a | 51.0 a | 96.3 a | 100.0 a |
| Glyphosate | 2592 | 50.0 a | 53.2 a | 100.6 a | 100.0 a |
| Glyphosate/glyphosate | 1296/1296 | 54.5 a | 58.5 a | 89.2 b | 100.0 a |
| Glyphosate + *B. subtilis* | 1296 + 42 | 55.7 a | 56.2 a | 106.7 a | 100.0 a |
| Glyphosate + *B. subtilis* | 2592 + 42 | 78.5 a | 53.8 b | 104.8 a | 100.0 a |
| Glyphosate + *B. subtilis*/Glyphosate + *B. subtilis* | 1296 + 42/1296 + 42 | 54.5 a | 61.3 a | 98.9 a | 100.0 a |
| CV (%) | | 19.50 | | 6.29 | |

DC = double check. For each response variable, means followed by different letters in the line differ from each other by the F Test ($p \leq 0.05$).

In contrast, for the treatment with the application of glyphosate plus *B. subtilis* BV02 (2596 + 42 g a.i. ha$^{-1}$), the behavior observed was an increase equivalent to 45.9% in the number of pods per plant compared to the respective double check (Table 4). Although

this behavior is not an expected response due to the use of this treatment in soybeans, this increase in the number of pods per plant may be related to the presence of the product based on *B. subtilis* BV02 in the composition.

For grain yield, differences were observed in only one case between treatments and the respective checks. A difference in the grain yield between the treatment and the check was observed only for the treatment with a sequential application of glyphosate/glyphosate (1296/1296 g a.i. ha$^{-1}$), with a reduction equivalent to 10.78%. Despite this, comparing the effect of treatments in which the sequential application of glyphosate was carried out at the same doses in which the percentage reduction in soybean yield was visualized, but in which the application was carried out in association with *B. subtilis* BV02, we highlight the occasion in which no reductions were observed in this variable compared to the respective check.

## 4. Discussion

The introduction of soybean cultivars with a tolerance to the application of glyphosate changed the scenario related to the common visualization of high intensities of injuries in plants due to the use of herbicides with other mechanisms of action applied in the post-emergence of conventional cultivars [17]. In general, the symptoms caused by the application of glyphosate in post-emergence in the soybean are characterized as mild, as long as the herbicide is positioned within the technical recommendations, with yellowing in the leaf blade (chlorosis) of the new trifoliate leaves emitted by the plants [18].

In contrast, when glyphosate is positioned outside the technical recommendations for the crop, with the use of doses higher than the maximum label insert, application in plants under abiotic stresses (e.g., drought), and associations with other active ingredients, there are risks of observing higher levels of injuries in soybean plants [5]. This is because in these situations, the plant may not be able to fully detoxify the glyphosate, or the injuries seen refer to secondary symptoms arising from the toxic action of the active ingredient (e.g., glyphosate chelating action) [19].

Given the context discussed above, even knowing the selectivity that glyphosate presents for tolerant soybean cultivars, it is essential to evaluate the intoxication of these plants subjected to the application of this active ingredient in post-emergence. Furthermore, in the hypotheses that make up the present work, assessing phytotoxicity allows us to measure whether *B. subtilis* BV02 has the potential to be used in the attenuation of visual injuries caused by the application of glyphosate on the crop. When compiling the results of the four phytotoxicity evaluations, some findings can be made: in general, the application of higher doses of glyphosate-based products provides greater intensity of injuries to the plants, and the sequential application with fractioned doses (2 × glyphosate-1296 g a.i. ha$^{-1}$) is more problematic than the use of the same dose in a single application (glyphosate-2592 g a.i. ha$^{-1}$); in addition, the association of *B. subtilis* BV02 with glyphosate does not result in major benefits in the attenuation of symptoms caused by this herbicide when applied in the post-emergence of the soybean crop.

Among the main symptoms observed in the soybean after glyphosate application, the occurrence of an intense yellowing of the treated and younger trifoliate leaves of the plants stands out [18]. In this context, the evaluation of the coloration of the leaves quickly after the application of glyphosate can help in the identification of possible deleterious effects. The results described for chlorophyll content demonstrate the problem related to the post-emergence application of glyphosate-based products in transgenic soybeans, since at higher doses, there was a reduction in the indirect variables related to these pigments. In this sense, the work achieves results similar to those already reported in the literature, clearly demonstrating the deleterious effects that glyphosate can have on photosynthetic pigments when applied in certain recommendations [5,7]. On the other hand, the results obtained in the present work demonstrate the potential of using *B. subtilis* BV02 in association with the glyphosate-based product to mitigate the negative effects of this herbicide on the variables related to the relative indices of chlorophyll.

*B. subtilis* has been reported to be able to regulate stomatal movements, and although it is most commonly associated with stomatal closure [20–23], and some studies also relate this microorganism to stomatal opening [24,25]. It should be noted that the studies that correlated *B. subtilis* with the induction of stomatal closure were carried out in plants subjected to drought [20,22,23] or to pathogens [21], stresses in which the decrease in gs is an important defense mechanism. Exposure to glyphosate, on the other hand, can stimulate stomatal opening [21,26], and the microorganism exacerbated this effect. These results indicate, therefore, that the response triggered by application with *B. subtilis* is variable and dependent on the plant species and on other factors that also influence plant metabolism, such as the environmental conditions to which the plants were submitted.

For chlorophyll a, changes in fluorescence parameters were not observed in any of the treatments, which indicates that the herbicide did not damage the proteins or components of PSII [27]. In addition, PSII characterizes the percentage of absorbed light that is effectively used in photochemical reactions, and similar results have already been observed in *Carthamus tinctoriu* exposed to glyphosate [28]. Benefits of the use of *B. amyloliquefaciens* in plants of *Medicago sativa* under drought stress have also been reported in the literature, since photosynthesis and antioxidant characteristics showed better values by inoculating the plants with this microorganism compared to those without [29].

The soybean crop consists of a species that is characterized by having high phenotypic plasticity, a characteristic that is related to the plant's ability to change its morphology and yield components to adapt them to the conditions imposed by the spatial arrangement of plants in the production area [30]. Despite the high phenotypic plasticity of this crop, there are situations in which reductions in the plant population can lead to lower grain yield, since these morphological changes to compensate for eventual failures in the soybean stand have a limited capacity. In this sense, for experiments that evaluate the selectivity of herbicides, it is necessary to measure the effects of these molecules on plant mortality, since reductions in this variable can result in a reduction in crop yield. Furthermore, among the main components of soybean yield, the final plant population is listed, presenting a direct relationship with the crop's grain yield [31]. Another variable that is particularly interesting is the plant height, since it influences certain growth parameters, such as plant lodging potential, or production losses in the mechanized harvesting operation, when the pods are at lower heights than the harvester's cutting platforms [6]. As already shown, differences in these variables were not seen in the present work.

Possible explanations for the low visualization of the effect of treatments on the morphological parameters of soybean plants are related to the good levels of rainfall observed before and after the application of treatments (Figure 1), a fact that may have contributed to a greater recovery of the crop as a result of the negative effects provided by the herbicide. Furthermore, adequate crop nutrition, through inoculation practices, aiming at nitrogen supply and fertilization may have contributed to a greater tolerance of soybean to the glyphosate-based product application. Despite this, increases in yield components and crop yield as a result of the application of products based on this microorganism to soybean have already been reported in the literature, but with a different strain [32]. The results of this work demonstrate that the addition of *B. subtilis* BV02 to the glyphosate application solution helped to attenuate the deleterious effects provided by glyphosate on the soybean, preventing losses in crop yield to be observed.

## 5. Conclusions

The single application of glyphosate (2592 g a.i. ha$^{-1}$) and the sequential application of this herbicide using, in both applications, a dose of 1296 g a.i. ha$^{-1}$ provides higher levels of intoxication to soybean plants, regardless of the application of *B. subtilis* BV02 in association with these treatments. The association of *B. subtilis* BV02 in both sequential applications of glyphosate prevented reductions in the values of the relative index of chlorophyll *b*, which was verified in comparison with the check without the application of herbicides. Furthermore, the association of *B. subtilis* BV02 to glyphosate (2592 g a.i. ha$^{-1}$)

meant that reductions in the values of relative chlorophyll *a* and *b* and the total index were not observed. The soybean yield was reduced when the plants were submitted to the sequential application of glyphosate (1296/1296 g a.i. ha$^{-1}$). The addition of *B. subtilis* BV02 to this treatment prevents losses in grain yield.

**Author Contributions:** Conceptualization, G.B.P.B. and E.S.F.; methodology, G.B.P.B., B.C.S.P. and F.d.S.F.; validation, B.C.S.P., L.L.-L. and L.F.d.S.; formal analysis, G.B.P.B., F.d.S.F. and M.d.F.S.; investigation, B.C.S.P., L.L.-L. and L.F.d.S.; data curation, G.B.P.B. and F.d.S.F.; writing—original draft preparation, G.B.P.B., E.S.F., M.d.F.S. and F.d.S.F.; writing—review and editing, all authors; visualization, G.B.P.B. and M.d.F.S.; supervision, G.B.P.B. and E.S.F.; funding acquisition, G.B.P.B. and E.S.F. All authors have read and agreed to the published version of the manuscript.

**Funding:** This research received no external funding and The APC was funded by Grupo Vittia (Brazil).

**Data Availability Statement:** Data are available from the corresponding author.

**Acknowledgments:** The authors thank the Universidade de Rio Verde for the infrastructure and logistic support to develop the present research.

**Conflicts of Interest:** The authors declare that the research was conducted in the absence of any commercial or financial relationships that could be construed as a potential conflict of interest.

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
