# Peer review of "Agronomic Performance of RR® Soybean Submitted to Glyphosate Application Associated with a Product Based on Bacillus subtilis"

_agronomy, doi:10.3390/agronomy12122940_

Round 1

Reviewer 1 Report (Previous Reviewer 4)

I feel authors could to significant changes in the manuscript to improve its quality. But I do not see any major changes in the revised version.

Author Response

Reviewer 2 Report (Previous Reviewer 3)

Corrections are done.

Author Response

Reviewer 3 Report (Previous Reviewer 2)

The authors made all suggested changes, looks good 

Author Response

Reviewer 4 Report (Previous Reviewer 1)

The introduction provides sufficient background and includes all relevant references to investigate the use of growth-promoting microorganisms to attenuate injuries caused by herbicides. In this case, the agronomic performance of RR® soybean submitted to post-emergence application of glyphosate isolated and in association with Bacillus subtilis. All the cited references are relevant to the research. The research design Is appropriate. Methods are adequately described. The results are clearly presented. The results support the conclusions.

Line 399:  PSII instead of ΦPSII

Line 401: Medicago instead of Mendicago

Round 2

Reviewer 1 Report (Previous Reviewer 4)

Manuscript looks fine.

This manuscript is a resubmission of an earlier submission. The following is a list of the peer review reports and author responses from that submission.

Round 1

Reviewer 1 Report

The introduction provides sufficient background and includes all relevant references to investigate the use of growth-promoting microorganisms to attenuate injuries caused by herbicides and in this case the agronomic performance of RR® soybean submitted to post-emergence application of glyphosate isolated and in association with Bacillus subtilis. All the cited references are relevant to the research. The research design Is appropriate. Methods are adequately described. The results are clearly presented. The conclusions are supported by the results.

The key is to see if this effect is also be seen in weeds, which is rather unwelcome.

Line 305-308: why don't we see the same EL value in the GLY D2 treatment?

Reviewer 2 Report

Expand RR when it appears first 

Line nos. 15-17 Please clearly mention the doses; Using 2x for multiple applications is confusing as the weed science community uses x for depicting times of field dose. 

Line o. 19 "this herbicide"? Which herbicide?

Line 21-22 Not clear; please rewrite

Line 63-64 Are the microorganisms used as plant growth regulators or biocontrol agents? Both are different. Please clarify.

 Line 69-64 Not sure how robust this hypothesis is; please include more references

Line 83 Please provide the soil type in English. 

Reviewer 3 Report

Only:

line 80: carry out instead installed

line: 398: B. amyloliquefaciens - in italics

Why 1 year field study only?

Reviewer 4 Report

1. Introduction must include and discuss previous studies. 

2. It is unclear why authors select Bacillus subtilis for this study. There are several biocontrol agents present.

3. The study lacks scientific motivation and novelty.

4. Authors do not demonstrate mechanistic aspects of the Bacillus subtilis in intoxication.

5. Authors need more biological experiments and data to support their claim. Visual scale and chlorophyll content is not enough to make such a claim. 

6. Visual examination and chlorophyll levels can vary due to several other reasons.

7. Figures quality need to be improved.